# Scaling up Image Segmentation across Data and Tasks

## Abstract

Traditional segmentation models, while effective in isolated tasks, often fail to generalize to more complex and open-ended segmentation problems, such as free-form, open-vocabulary, and in-the-wild scenarios. To bridge this gap, we propose to scale up image segmentation across diverse datasets and tasks such that the knowledge across different tasks and datasets can be integrated while improving the generalization ability. QueryMeldNet, a novel segmentation framework, is introduced and designed to scale seamlessly across both data size and task diversity. It is built upon a dynamic object query mechanism called query meld, which fuses different types of queries using cross-attention. This hybrid approach enables the model to balance between instance- and stuff-level segmentation, providing enhanced scalability for handling diverse object types. We further enhance scalability by leveraging synthetic data-generating segmentation masks and captions for pixel-level and open-vocabulary tasks-drastically reducing the need for costly human annotations. By training on multiple datasets and tasks at scale, QueryMeldNet continuously improves performance as the volume and diversity of data and tasks increase. It exhibits strong generalization capabilities, boosting performance in open-set segmentation tasks SeginW by 7 points. These advancements mark a key step toward universal, scalable segmentation models capable of addressing the demands of real-world applications.

## 1 Introduction

Image segmentation is an important computer vision research direction with the goal of partitioning an image into discrete groups of pixels. This field encompasses various training tasks, including semantic segmentation, instance segmentation, panoptic segmentation, foreground/background segmentation, and referring segmentation, etc. The objective of a universal image segmentation model is to exhibit robust generalization capabilities, performing effectively in real-world diverse segmentation applications, such as open-vocabulary, free-form and in-the-wild segmentation requirement Xu et al. (2023); Liu et al. (2023); Zou et al. (2023a). To achieve that, such a model is expected to be trainable jointly across any segmentation datasets and tasks *at scale* such that the knowledge across different tasks and datasets can be integrated. This integration is essential for improving performance on complex, real-world problems, particularly when larger and more diverse datasets are available. We say that a segmentation model is *scalable* if it can effectively improve with the increase in both dataset size and task diversity. A scalable model can continuously evolve by leveraging existing and future datasets, without requiring frequent redesign or retraining, making development more efficient. In this way, simply gathering more diverse data can naturally enhance the model's capabilities.

Despite these benefits, numerous prior works were explored on specific tasks or datasets in isolation He et al. (2017); Chen et al. (2017; 2019); Ronneberger et al. (2015); Long et al. (2015); Xiong et al. (2019); Cheng et al. (2020). While these models have achieved significant success in their respective areas, they often struggle to generalize to real-world scenarios, where versatility and adaptability are critical. The limitations of task-specific models raise a key question: Can we design a model that scales effectively across both tasks and datasets while improving generalization in diverse, real-world applications?

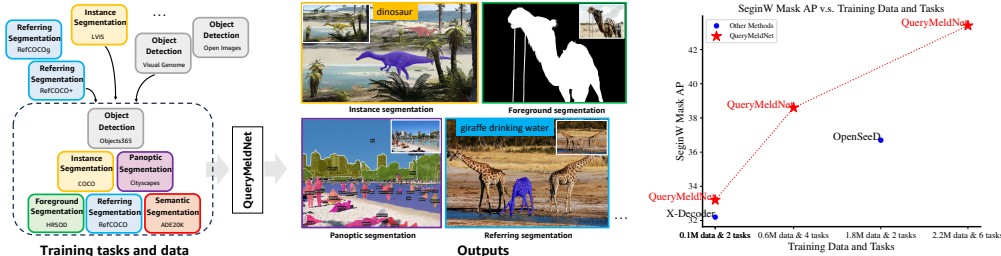

Figure 1: QueryMeldNet, a scalable segmentation model, is designed to train across a wide range of datasets and segmentation tasks, including both existing and newly introduced ones. The model supports open-vocabulary inference and excels in handling multiple segmentation tasks simultaneously, such as instance, panoptic, semantic, and referring segmentation. The graph demonstrates the model's strong generalization capabilities, as indicated by its performance improvements on the SeginW benchmark, which scales efficiently with increasing amounts of training data and tasks.

Several recent efforts have aimed to scale up segmentation training tasks and datasets by exploring unified frameworks, seeking to address the joint training of multiple tasks and datasets, summarized in Table 1. However, these existing works possess certain inherent limitations, far from achieving true scalability across both tasks and datasets. Some of these works have made progress in dataset scalability but remain restricted to a single task Lambert et al. (2020); Kim et al. (2022a). Others Jain et al. (2023); Zhang et al. (2023) have demonstrated limited task scalability—addressing only specific tasks such as semantic, instance, or panoptic segmentation—but cannot generalize across datasets with different class structures. There is few attempt for both datasets and tasks scalability Gu et al. (2023); Zou et al. (2023a). X-Decoder Zou et al. (2023a) offers a promising solution with its learnable queries for jointly training on tasks and datasets. Nevertheless, its subpar performance in instance-level segmentation reveals shortcomings in its architecture, indicating that its task scalability is still constrained.

In this work, we conduct an in-depth analysis and identify a key limitation preventing effective scalability: the design of object queries, a fundamental component in transformer-based segmentation models. The learnable queries used in X-Decoder have shown promising results for semantic (stuff) segmentation but struggle with instance (thing) segmentation[1]. To address this issue, we draw inspiration from the success of conditional queries in object detection Liu et al. (2022a); Li et al. (2022); Zhu et al. (2020); Zhang et al. (2022a) and introduce them to enhance X-Decoder's ability in instance-level segmentation and broaden its scalability across both tasks and datasets. However, while conditional queries excel at instance objects, they perform poorly with stuff objects. To harmonize the strengths of both query types, we propose a novel object query mechanism called *query meld*. This approach seamlessly melds learnable queries and conditional queries with a qual-query cross attention mechanism. It enables sample and object-wise dynamic query selection, opposite to traditional rigid assignment Rana et al. (2023); Athar et al. (2023); Zhang et al. (2023), and hierarchical and interactive feature representation which improve the model's ability to handle diverse object types, enabling scalability across various tasks and datasets.

Building on this foundation, we introduce a scalable segmentation architecture called QueryMeldNet. QueryMeldNet can be trained on many different segmentation tasks and datasets at scale, as shown in Figure 1, without being limited to specific datasets Kim et al. (2022a); Lambert et al. (2020); Zhou et al. (2023) or tasks Jain et al. (2023); Zhang et al. (2021) as previous works. A key advantage of QueryMeldNet's scalable design is its ability to continuously improve segmentation performance by training on a wide variety of existing datasets and tasks. We demonstrate that scaling up both the volume of training data and diversity of tasks consistently enhances the model's segmentation capabilities, particularly for real-world, free-form open-set segmentation tasks. As shown in Figure 1 (right), when we scale the data and tasks from 0.1M to 0.6M and include more diverse tasks, the open-set segmentation mask AP performance on the SeginW benchmark Zou et al. (2023a) improves from 33.2 to 38.6. While current public datasets provide a good starting point, we are eager to explore the limits of the model's generalization capabilities by utilizing even more

---

[1]The term "thing" (referring to countable objects, usually in the foreground) and "stuff" (referring non-object, uncountable elements, often in the background) are frequently employed to make a distinction between objects with clearly defined geometry and quantifiability, such as people, dogs, and surfaces or areas lacking a fixed geometry, primarily recognized by their texture or material, like sky, road Kirillov et al. (2019).

Table 1: Summary of data and task scalability of related image segmentation works. Unlike previous works that are only scalale to specific datasets or limited tasks, QueryMeldNet overcomes these constraints by enabling joint data and task scalability.

| | Data Scalability | Task Scalability | | | | | |
| --- | --- | --- | --- | --- | --- | --- | --- |
| | | Instance | Semantic | Panoptic | Referring | Foreground | Detection |
| MSeg Lambert et al. (2020) | ✓ | | ✓ | | | | |
| UniSeg Kim et al. (2022a) | ✓ | | ✓ | | | | |
| OneFormer Jain et al. (2023) | | ✓ | ✓ | ✓ | | | |
| OpenSeeD Zhang et al. (2023) | | ✓ | ✓ | ✓ | | | ✓ |
| X-Decoder Zou et al. (2023a) | ✓ | ✓ | ✓ | ✓ | ✓ | | |
| DataSeg Gu et al. (2023) | ✓ | | ✓ | ✓ | | | ✓ |
| **Our QueryMeldNet** | ✓ | ✓ | ✓ | ✓ | ✓ | ✓ | ✓ |

diverse segmentation data. However, human annotation for segmentation is usually expensive, *e.g.*, requiring a few minutes to annotate a single COCO image. To circumvent this data limitation, we propose to harness synthetic data, *i.e.*, synthetic segmentation masks for pixel-level segmentation and synthetic segment captions for open-vocabulary semantic alignment. This is feasible as some recent models can already generate impressive synthetic segmentation masks Kirillov et al. (2023); Ke et al. (2023) and object-level captions Wang et al. (2022b); Zhang et al. (2022b), and the synthetic data has been proven helpful for model improvement Cho et al. (2023); Gao et al. (2022). With the low cost of generating synthetic data, we can easily scale up training. Incorporating synthetic data not only mitigates the challenge of data scarcity but also strengthens the model's robustness and semantic understanding. By further scaling with synthetic data, QueryMeldNet pushes its performance even higher, reaching 43.2, an additional improvement of 4.6 points. These advancements represent a significant step toward developing a scalable and highly generalized image segmentation model.

Overall, this paper has three major contributions. First, we introduce QueryMeldNet, a scalable segmentation architecture that can be jointly trained and evaluated on any segmentation task and dataset, breaking the constraints of task or dataset specific models, making it possible to scale up image segmentation model across both datasets and tasks. Second, we demonstrate that scaling up the model across diverse tasks and datasets consistently enhances its generalization ability. Third, by incorporating synthetic data to further scale up the model, QueryMeldNet achieves state-of-the-art performance on multiple open-set segmentation benchmarks.

## 2 RELATED WORK

**Generic segmentation** Given an input image, the goal of image segmentation is to output a group of masks with class predictions. According to the scope of class labels and masks, image segmentation can be divided into three major tasks, semantic, instance and panoptic segmentation Li et al. (2023c). In the past, many task or dataset specialized models have been proposed, and they can be trained and do inference only on a single task and dataset, including Mask R-CNN He et al. (2017), Cascade Mask R-CNN Cai & Vasconcelos (2019), HTC Chen et al. (2019) on instance segmentation, FCN Long et al. (2015), U-Net Ronneberger et al. (2015), DeepLab Chen et al. (2017) on semantic segmentation, UPSnet Xiong et al. (2019), Panoptic-DeepLab Cheng et al. (2020) on panoptic segmentation.

**Scalable segmentation models** Most early unified segmentation models lack scalability because their architectures need modifications to accommodate different datasets and tasks Cheng et al. (2021; 2022); Li et al. (2023a). For instance, in Mask DINO Li et al. (2023a), training on semantic segmentation requires a one-stage encoder-decoder architecture, whereas instance and panoptic segmentation demand a two-stage approach. This inconsistency limits scalability across tasks. Some models achieve partial scalability, either for tasks Jain et al. (2023); Zhang et al. (2021); Qin et al. (2023) or datasets Kim et al. (2022a); Lambert et al. (2020); Zhou et al. (2023), but not for both. For example, OneFormer Jain et al. (2023) and OpenSeeD Zhang et al. (2023) handle task scalability within instance/semantic/panoptic segmentation but struggle with dataset scalability. OneFormer lacks the ability to unify class spaces across datasets, while OpenSeeD requires additional stuff/thing annotations, which are impractical for most datasets. Few models attempt to address both data and task scalability. X-Decoder Zou et al. (2023a) and DaTaSeg Gu et al. (2023) offer a sub-optimal solution by relying on learnable queries, but they exhibit decreased performance in instance segmentation. To the best of our knowledge, no segmentation model currently supports both data and

task scalability while performing well across tasks and showing good generalization ability. In this work, QueryMeldNet aims to solve this challenge. Table 1 compares each method.

**Using synthetic data for stronger model** Cho et al. (2023) uses an image captioning model to generate pseudo captions on the cropped object regions for object detection, but it neglects the context information during the object caption generation. Pseudo bounding boxes are also leveraged to expand the training data size Gao et al. (2022). For image segmentation, PseudoSeg Zou et al. (2020) designs a one-stage framework to generate pseudo masks from unlabeled data or image-level labeled data for semantic segmentation. Another line producing and applying pseudo labels to improve the model is under the teacher-student semi-supervised learning framework Chen et al. (2021); Wang et al. (2022d); Liu et al. (2022b). OpenSeeD Zhang et al. (2023) also uses a pseudo mask generator decoding from bounding boxes during training. However, we argue that all these on-the-fly pseudo data generation methods will increase the training cost. In our work, inspired by the recent segmentation models that can generate high-quality mask predictions Kirillov et al. (2023); Ke et al. (2023) and have been shown to be a good pseudo label generator Jiang & Yang (2023); Chen et al. (2023), we generate the synthetic data offline, which will be used during training with no difference from ground truth.

## 3 METHOD

In this section, we first present an overview of the QueryMeldNet architecture. We then introduce the novel query meld mechanism, a key component that drives effective scalability within the architecture. Next, we explain how QueryMeldNet scales across both data and tasks. Finally, we outline our efforts to further enhance scalability using synthetic data.

### 3.1 QUERYMELDNET ARCHITECTURE

Figure 2 shows the architecture of the proposed QueryMeldNet. It has four major components, image and text encoder, and segmentation encoder and decoder. The image encoder encodes an input image to multi-scale image features, and the text encoder encodes the text query to obtain its semantic embedding. The multi-scale image features are forwarded to the segmentation encoder for further refinement. Next, the segmentation decoder takes numbers of object queries and attends the image features with query meld mechanism to predict the final class, bounding box, and segment mask.

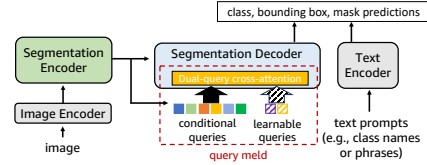

Figure 2: The overview of QueryMeldNet architecture. The model takes an image and a list of textual language prompts as input and outputs their corresponding localized segment masks.

### 3.2 QUERY DESIGN FOR SCALABLE SEGMENTATION

Object query is a key component in transformer-based object detection and segmentation models, and has attracted much attention from the community Liu et al. (2022a); Li et al. (2022); Zhu et al. (2020); Zhang et al. (2022a); Wang et al. (2022c); Meng et al. (2021). In this section, we first review the mostly common learnable object query strategy in segmentation architectures and introduce our new query meld mechanism.

**Learnable query** relies on a single set of object queries trained from scratch which interact with the image features to encode object location and class information (illustrated in Figure 3 (a)). Due to its simplicity, this approach has been widely adopted in the object detection and segmentation literature Wang et al. (2022a); Zou et al. (2023a); Cheng et al. (2021); Gu et al. (2023). For example, X-Decoder Zou et al. (2023a) uses learnable queries in an attempt to achieve data and task scalability. However, several studies have demonstrated that learnable queries perform suboptimally in object detection Zhu et al. (2020); Liu et al. (2022a); Li et al. (2022). Our experiments reveal similar findings in image segmentation: while learnable queries perform well for semantic segmentation, they fall short in instance-level tasks such as instance segmentation. As shown in Table 2, there is a noticeable performance gap compared to more advanced query designs. This limitation hampers X-Decoder's ability to scale up across diverse and complex data and tasks, restricting its broader scalability.

To address the shortcomings of learnable queries in instance-level segmentation, we explore more advanced query designs that have proven successful in object detection. One such approach is the **conditional query** Liu et al. (2022a); Li et al. (2022); Zhu et al. (2020); Zhang et al. (2022a), initially proposed in Zhu et al. (2020) and further refined in Liu et al. (2022a); Li et al. (2022); Zhang et al. (2022a). Conditional queries aim to mimic the proposal generation mechanism found in traditional two-stage object detection frameworks Ren et al. (2015), but adapted for transformer-based detectors. Unlike learnable queries, which are independently trained, conditional queries are derived directly from the transformer encoder, as illustrated in Figure 3 (b). The transformer encoder is trained to predict region proposals, from which high-confidence proposals are selected and fed into the transformer decoder as object queries for final predictions, such as bounding boxes or segmentation masks.

Conditional queries align more closely with the objects likely to be present in an image and have consistently demonstrated superior performance in object detection tasks Zhu et al. (2020). However, our experiments reveal that this strategy does not universally benefit all segmentation tasks. As shown in Table 2, the performance on semantic segmentation is significantly worse compared to learnable queries. This is because, in semantic segmentation, many classes (often referred to as "stuff" classes) represent background regions with undefined shapes and spatial extents. Conditional queries, derived from local image features, struggle to capture these characteristics effectively, leading to suboptimal results. This is different from learnable query that is learned from scratch, not conditional on an encoder output that usually derived from a local patch feature. Since stuff classes are prevalent in real-world datasets, relying solely on conditional queries also limits the scalability of models across diverse tasks and datasets.

Both learnable and conditional queries have their respective strengths: learnable queries excel at handling large, amorphous background regions, while conditional queries specialize in capturing local, instance-level features. However, their individual limitations restrict their scalability across a wider range of datasets and tasks. This raises a simple yet powerful idea: can we combine the strengths of both to enhance scalability? Following this line of thinking, we propose a **query meld** strategy (Figure 3 (c)). In this approach, the object query set consists of both learnable and conditional queries, which interact with each other through a deep fusion mechanism via dual-query cross-attention. For loss computation, Hungarian matching is applied across all object queries, without differentiating between query types, allowing the model to seamlessly integrate both types of queries for improved scalability across diverse segmentation datasets and tasks.

With dual-query cross-attention mechanism, the query meld seamlessly integrates learnable queries with conditional queries, offering several key advantages. First, dynamic query selection. Without rigid assignment, two types of queries can dynamically choose their preferred objects to detect for each example. And since they are complementary each other for global background feature and local instance feature, this property broadens the scope of the trainable dataset and tasks and therefore improves the scalability of the model. Second, hierarchical and interactive feature representation. Dual-query cross-attention can lead to a hierarchical feature representation where learnable queries capture the overall structure and semantics of the objects in the scene. On the other hand, conditional queries refine these global features by attending to specific parts of the image. This interaction allows the model to dynamically adjust focus, using conditional queries to zero in on hard-to-segment objects while still retaining the global understanding provided by learnable queries. This can improve the model's ability to handle both coarse and fine segmentation tasks. For complex objects or occluded regions, query meld could also provide complementary perspectives on the same object. Overall, the introduction of query meld enables the architecture to handle a broader range of segmentation tasks and data in a flexible manner. The system can dynamically prioritize either query type based on the complexity and nature of the task, benefiting better generalization ability of the model. We will see the benefits in experiment section.

### 3.3 SCALABLE SEGMENTATION ACROSS DATA AND TASKS

Under our QueryMeldNet architecture, we are ready to scale up image segmentation both for datasets and tasks. This thanks to a neat and unified input data format of training QueryMeld-Net. For any segmentation datasets of different tasks, the training set can always be reformulated to a unified format $\mathcal{D} = \{(\mathbf{x}_i, \mathbf{y}_i)\}_{i=1}^N$ where $\mathbf{x}_i$ is the image and $\mathbf{y}_i = \{(c_j, \mathbf{b}_j, \mathbf{m}_j)\}_{j=1}^B$ its $B$ annotations. $(c_j, \mathbf{b}_j, \mathbf{m}_j)$ is a triplet depicts a single mask annotation on the image. $c_j$ is the se-

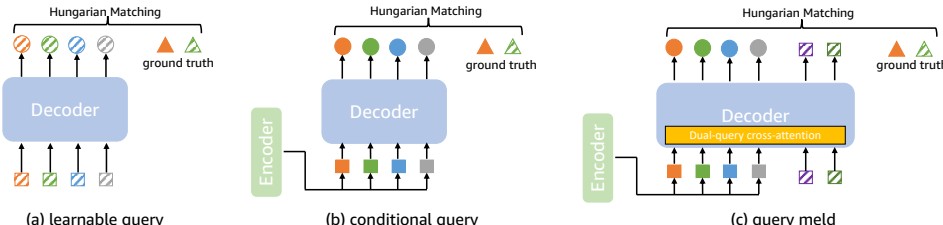

Figure 3: The comparison of different query strategies. Square with diagonal slashes: learnable query; solid square: conditional query; circle with slashes: query embedding of learnable queries; solid circle: query embedding of conditional queries; triangle with slashes: ground truth of stuff class; solid triangle: ground truth of thing classes. (a) learnable query is learned from scratch. (b) conditional query is derived and selected from encoder. (c) query meld fuses both types of queries by dual-query cross attention.

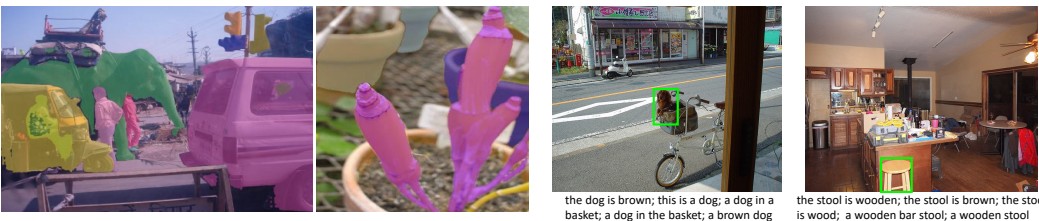

Figure 4: Synthetic data visualization. Left: synthetic masks by SAM; Right: synthetic captions by OFA-akin model.

mantic class label (*e.g.*, "apple", "road" for semantic/instance/panoptic/foreground segmentation), or a text description (*e.g.*, "a person wearing a red shirt" for referring segmentation), to describe the semantic information characterized with the binary mask region $\mathbf{m}_j$. $\mathbf{b}_j$ is the bounding box annotation of this region which can be derived from the mask annotation. Note that $c_j$ could be any natural language description without demanding extra annotation and the training data is fed to the model without extra assignment or discrimination. This is unlike some literature Rana et al. (2023); Athar et al. (2023); Zhang et al. (2023); Li et al. (2024) that make the hard assignment for each query to different tasks or classes, for instance, OpenSeeD Zhang et al. (2023) that requires extra stuff/things discrimination annotations for each class. This limitation restricts its dataset and task scalability because such annotation is not available for most of public datasets like Objects365 Shao et al. (2019), OpenImages Kuznetsova et al. (2020), Visual Genome Krishna et al. (2017) since there is no clear boundary between stuff and thing classes. For example, "window" and "table" classes are labeled as thing in ADE20K Zhou et al. (2017) but as stuff in COCO Lin et al. (2014). For some segmentation tasks like referring segmentation, it even can not classify its free-form annotation into stuff/things.

The whole model thus can be trained with loss function as follows (for clarity, we omit the weight for each loss term),

$$\mathcal{L}=\sum_{(\mathbf{x}_i,\mathbf{y}_i)\in\mathcal{D}}\sum_{(c_j,\mathbf{b}_j,\mathbf{m}_j)\in\mathbf{y}_i}\mathcal{L}_c(\mathbf{P}^c(\mathbf{x}_i),\mathbf{H}(c_j))+\mathcal{L}_b(\mathbf{P}^b(\mathbf{x}_i),\mathbf{b}_j)+\mathcal{L}_m(\mathbf{P}^m(\mathbf{x}_i),\mathbf{m}_j), \quad (1)$$

where $\mathcal{L}_c$, $\mathcal{L}_b$, $\mathcal{L}_m$ are the class, bounding box (bbox) and mask loss, respectively. They are applied to class, bbox and segment mask embeddings, $\mathbf{P}^c, \mathbf{P}^b, \mathbf{P}^m$, from the decoder outputs and text embedding $\mathbf{H}$, for supervision. The class loss is the focal loss Lin et al. (2017) applied on the dot-product between the class embedding and text embedding. The bbox loss is generalized IoU and L1 loss Rezatofighi et al. (2019) between the bounding box embedding and ground truth. The mask loss is calculated with generalized dice loss Sudre et al. (2017) on the mask prediction which is derived from the mask embedding and a pixel encoder. Since all semantic class labels are in the form of textual description, and will be encoded by the text encoder, as shown in Figure 2. So the model is capable of dealing open-vocabulary and free-form scenarios and there is no need for sophisticated label space alignment across datasets with different semantic labels. The unified data and training format of QueryMeldNet and soft constraint on the annotation of training data lead QueryMeldNet is scalable to wider diverse datasets and tasks.

### 3.4 SCALABILTY TO MORE DATA AND TASKS

To push the boundaries of the scalable image segmentation model, we aim to scale it up to encompass more diverse datasets and tasks. However, the sizes of well curated segmentation datasets are

Table 2: The performance comparison of different query strategies.

| Query strategy | Scalability | | Training data | Instance COCO Mask AP | Panoptic COCO PQ | Semantic ADE mIoU | Open-vocabulary SeginW Mask AP |
|---|---|---|---|---|---|---|---|
| | #Dataset | #Task | | | | | |
| learnable | 2 | 2 | COCO pano+ADE sem | 48.1 | 54.3 | 50.4 | 27.8 |
| | 4 | 4 | COCO pano+ADE sem+VG+refer | 48.6 | 54.1 | 50.1 | 32.1 |
| conditional | 2 | 2 | COCO pano+ADE sem | 49.8 | 56.5 | 43.2 | 29.4 |
| | 4 | 4 | COCO pano+ADE sem+VG+refer | 49.5 | 56.2 | 43.9 | 34.7 |
| query meld | 2 | 2 | COCO pano+ADE sem | 49.6(+1.5) | 56.5(+2.2) | 51.7(+8.5) | 30.6(+2.4) |
| | 4 | 4 | COCO pano+ADE sem+VG+refer | 49.9(+1.3) | 56.8(+2.7) | 52.1(+8.2) | 38.4(+6.3) |

usually relatively small [2] because pixel-wise mask annotation is expensive, which poses a significant limitation in exploring the full potential of scalability. To circumvent this challenge, we propose to use synthetic data, which is cheap to generate, easy to scale up and has been proven effective to strengthen the model, for instance, in object detection Cho et al. (2023); Gao et al. (2022) image captioning Davide et al. (2023). Given that some recent models can generate high-quality synthetic segmentation masks (e.g SAM Kirillov et al. (2023)) and synthetic captions (*e.g.*, OFA Wang et al. (2022b), GLIPv2 Zhang et al. (2022b)), we believe that the synthetic segmentation data can play a crucial role in exploring the scalability of our model. In this work, we leverage two types of synthetic data to expand both the training set and the range of tasks.

**Synthetic segmentation mask**: Instead of generating synthetic segmentation masks directly on unlabeled image, it is a much easier task to segment the mask given an object bounding box because some recent works have shown that they are pretty good at this task Kirillov et al. (2023); Ke et al. (2023); Zou et al. (2023b). The size of object detection dataset is usually more than dozen times larger than that of segmentation, *e.g.*, Objects365 Shao et al. (2019) of 1.7M images v.s. COCO Lin et al. (2014) of 120K images. With the generated synthetic masks, we can convert every object detection dataset to a segmentation dataset to have more diverse training data.

**Synthetic segmentation caption**: The standard segmentation/detection datasets usually lack rich textual descriptions, *e.g.*, 80 fixed category names for COCO. This is a big challenge for open-vocabulary segmentation model, especially for the task of referring segmentation. The widely used referring segmentation datasets are RefCOCO, RefCOCO+ and RefCOCOg as well as Ref-Clef Kazemzadeh et al. (2014); Yu et al. (2016), whose combination has only about 50K images. The reason to this small dataset size is because annotating a caption description to every individual object segment is expensive. In order to enrich the semantic information of the training data and improve the generalization ability of the model, we train a OFA-akin Wang et al. (2022b) model on the task of object captioning, *i.e.*, generating synthetic caption for each object given the bounding box. With this object captioning model, we generate five synthetic captions with the highest confidences for each object, and use them to expand the training data size. One of the synthetic captions is randomly selected per object at each training iteration.

## 4 EXPERIMENTS

To verify the dataset and task scalabilty of QueryMeldNet, we experiment on a variety of datasets proposed for different tasks: COCO Lin et al. (2014) and ADE20K Zhou et al. (2017) for comprehensive semantic/instance/panoptic annotations; LVIS Gupta et al. (2019) for instance segmentation; RefCOCO, RefCOCO+, RefCOCOg Kazemzadeh et al. (2014); Yu et al. (2016) for referring segmentation; HRSOD Zeng et al. (2019), DIS Qin et al. (2022), and other five datasets Cheng et al. (2015); Mansilla & Miranda (2016); Liew et al. (2021); Xie et al. (2022); Wang et al. (2017) for foreground segmentation; Objects365 Shao et al. (2019) and Visual Genome Krishna et al. (2017) for object detection. In addition, we generate synthetic captions on COCO, denoted as "COCO-syn" for referring segmentation. We also create synthetic masks for Visual Genome and Objects365, denoted as "Objects365-syn-m" for instance segmentation, and further generate synthetic captions on Objects365 for referring segmentation, "Objects365-syn".

To validate the real-world generalization ability of the model, several datasets or benchmarks are employed. Pascal Context Mottaghi et al. (2014) and BDD Yu et al. (2018) are used for open-set evaluation. SeginW benchmark which has 25 datasets is used for open-vocabulary in-the-wild seg-

---

[2] Although SA-1B Kirillov et al. (2023) is large, it relies on machine predictions and does not have semantic labels.

| Objects predicted by conditional queries | | | Objects predicted by learnable queries | | |
|---|---|---|---|---|---|
| Input image | Prompt | Output mask | Input image | Prompt | Output mask |
| | 'sky' | | | 'table' | |
| | 'floor' | | | 'car' | |

Figure 5: The counter prediction of examples by query meld. Left: the stuff objects are predicted with conditional queries instead of learnable queries; Right: the thing objects are predicted with learnable queries instead of conditional queries.

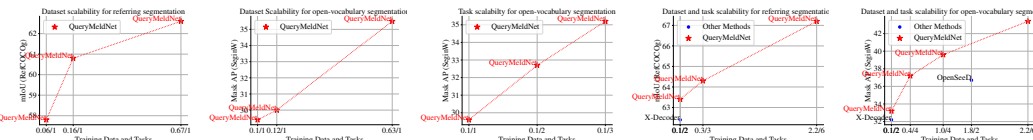

Figure 6: **The performance improvement with data and task scaling up**. The open-vocabulary (Mask AP of SeginW) and free-form segmentation (mIoU of RefCOCOg) ability keeps increasing with dataset and task scalability (the size of the training data (M)/ number of different training tasks). From left to right: only scaling up dataset for referring segmentation, only scaling up dataset for open-vocabulary segmentation, only scaling up task for open-vocabulary segmentation, scaling up both dataset and task for referring segmentation and scaling up both dataset and task for open-vocabulary segmentation.

mentation evaluation Zou et al. (2023a). RefCOCOg Yu et al. (2016) is used for free-form referring segmentation. We use mIoU as the evaluation metric for semantic and referring segmentation, Mask AP for instance segmentation, PQ Kirillov et al. (2019) for panoptic segmentation, following Li et al. (2023a); Zou et al. (2023a); Zhang et al. (2023); Kim et al. (2022b). The hyperparameters of the architecture and training follow Mask DINO Li et al. (2023a). The pretrained Swin Transformer Liu et al. (2021) and CLIP language encoder Radford et al. (2021) are adopt as the vision and text encoder, respectively, but it is noted that any vision or language backbone encoders can be used by QueryMeldNet. The query meld set consists of 100 learnable and 300 conditional queries, following some popular settings Li et al. (2023a); Zhang et al. (2023). For more details, please refer to the supplementary materials.

### 4.1 QUERY ABLATION

We begin by comparing three query strategies when used for scalable image segmentation. The model is scaled up to both datasets and tasks at two scales: (1) "two datasets and two tasks" where the training set comprises COCO with panoptic segmentation annotations ("COCO pano") and ADE20K with semantic segmentation annotations ("ADE sem"); (2) "four datasets and four tasks" where we add two additional training sets and tasks, Visual Genome with instance segmentation ("VG") and referring segmentation RefCOCO/RefCOCO+/RefCOCOg ("refer"). The evaluation uses ADE and COCO for closed-set performance, while SeginW is utilized to assess the open-set generalization capabilities of the models.

The query meld strategy is compared against the two other strategies, all using a total of 400 queries. As shown in Table 2, across both scaling scenarios, the learnable query exhibits weak performance on instance-level segmentation tasks, with a notable drop of around 2 points on COCO and SeginW. Even more significant, the conditional query shows a degradation of over 7 points in semantic segmentation performance (mIoU) on ADE. These results suggest that neither of the individual query strategies is an optimal choice for scalable image segmentation. In contrast, the query meld demonstrates superior performance across all evaluation tasks, highlighting its scalability to diverse tasks and datasets without suffering performance loss. Moreover, query meld exhibits stronger generalization ability, as evidenced by substantial performance improvements on SeginW, driven by its dual-query cross-attention mechanism.

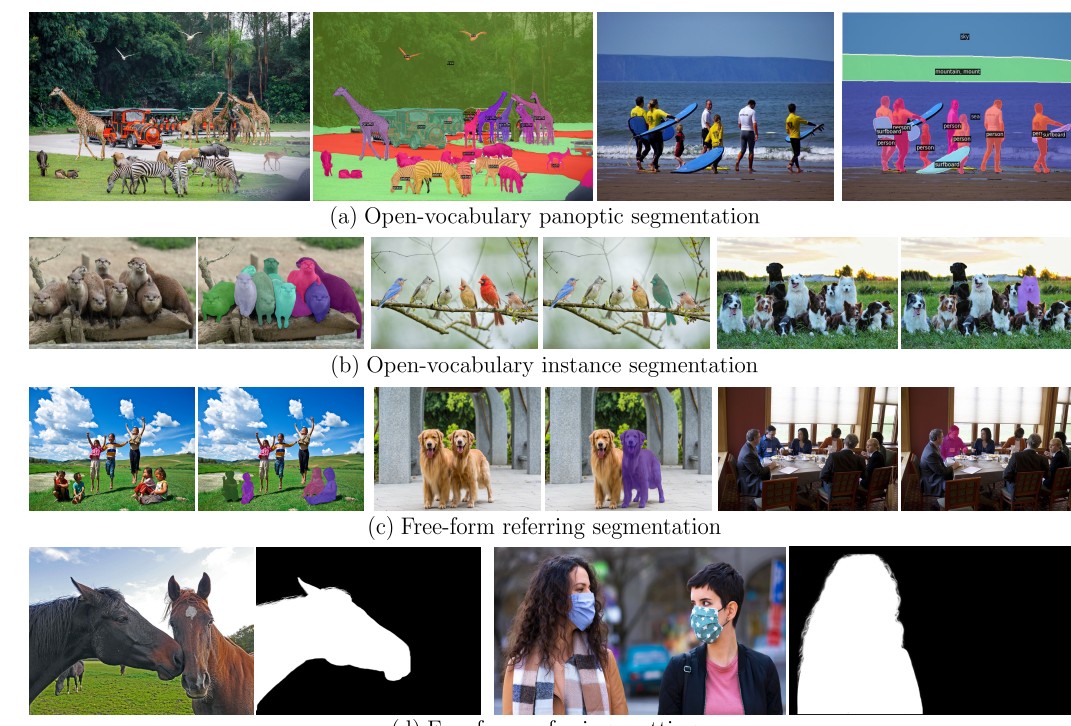

(a) Open-vocabulary panoptic segmentation

(b) Open-vocabulary instance segmentation

(c) Free-form referring segmentation

(d) Free-form referring matting

Figure 7: **Qualitative results of QueryMeldNet on each tasks**. For every pair of images, the left is the input image and the right is the prediction. The text prompts for the three examples in (b) are "otter", "Cardinal" and "Samoyed"; "children sitting in the grass", "right Golden Retriever", "person wearing a blue shirt" in (c) and the two prompts for (d) are "left horse" and "woman wearing a blue mask".

The superior performance of the query meld stems from its sample-wise dynamic query selection mechanism. We analyze the ratio of thing and stuff objects predicted by the conditional and learnable queries, respectively. Thing objects typically correspond to foreground instances, such as "person" or "book", while stuff objects generally represent background regions like "sky" or "road". On COCO, we find that conditional queries capture $99.6\%$ of thing objects, while learnable queries detect $53.3\%$ of stuff objects. A similar trend is observed in ADE panoptic segmentation, with conditional queries accounting for $99.8\%$ of thing objects and learnable queries handling $61.4\%$ of stuff objects. This suggests that, in most cases, thing objects are predicted by conditional queries, whereas stuff objects are handled by learnable queries. However, this is not always the case. Figure 5 illustrates counterexamples where, despite "sky" and "floor" being classified as stuff classes, conditional queries are used because these features behave more like local instances in the images. Similarly, in images containing "table" and "car", which are typically thing classes, learnable queries are triggered since these objects appear more as background features. These findings demonstrate that query selection in the query meld is dynamic and adaptive to each image, contrasting with some approaches in the literature Rana et al. (2023); Athar et al. (2023); Zhang et al. (2023) that rely on hard assignments based on classes or tasks, limiting scalability.

### 4.2 ABLATION ON DATA AND TASK SCALING UP

We next verify the scalability of QueryMeldNet across both datasets and tasks. The left two figures in Figure 6 demonstrate the model's dataset scalability. In the first figure, we evaluate the model on referring segmentation tasks, starting with training on RefCOCO/+/g datasets. By scaling up the training set to include additional COCO-sync data, the performance on RefCOCOg validation set improves from 57.8 to 60.8. Further scaling up to include $30\%$ of Objects365-syn dataset increases the performance to 62.6. A similar trend is observed for open-vocabulary tasks when the model is trained on instance segmentation and the dataset is scaled from COCO to COCO+ADE, and then to COCO+ADE+$30\%$ of Objects365-syn-m, as shown in the second figure. The middle figure illustrates task scalability. Training on a fixed 100K COCO images, we progressively scale the tasks from panoptic segmentation to include instance segmentation and referring segmentation

with synthetic description. The open-vocabulary performance increases steadily from 29.6 to 32.7 and then to 35.2. The last two figures validate the simultaneous scalability of both datasets and tasks. When scaling up both dimensions together, the referring and open-vocabulary segmentation tasks show consistent improvements. Notably, compared to the non-scalable OpenSeeD framework, which cannot benefit from additional training resources, QueryMeldNet demonstrates significant advantages. Furthermore, X-Decoder, due to its suboptimal learnable query strategy, underperforms QueryMeldNet on the same datasets and tasks.

### 4.3 COMPARISON WITH THE STATE-OF-THE-ART

We scale up our model with a larger set of datasets and tasks. We train it on around 2.2M distinct images examples from COCO, LVIS, Visual Genome, Objects365, Re-fCOCO/+/g and several foreground datasets and 57M mask annotations on six tasks (instance/semantic/panoptic/referring/foreground segmentation and object detection). The comparison is conducted on various open-set segmentation benchmarks considering open-set evaluation stands as a critical metric for assessing the generalization ability of a model, providing insights into its adaptability and performance in real-world applications. We evaluate the zero-shot performance on ADE20K for panoptic/semantic/instance segmentation, Pascal Context 59 (PC-59) with 59 common classes and PC-459 with full 459 classes Mottaghi et al. (2014) for semantic segmentation, and BDD Yu et al. (2018) for panoptic segmentation. The results are presented in Table 3. QueryMeldNet improves the state-of-the-art open-vocabulary segmentation on each benchmark. In order to further evaluate the generalization ability of QueryMeldNet, we evaluate it on the in-the-wild benchmark SeginW Zou et al. (2023a). The evaluation is conducted under the zero-shot setting. The comparison results are given in the last column, where our model has a significant improvement (7.3 points) over the prior art. This benefits from its scalability so that more diverse data and task are included during training, leading better knowledge integration and fusion, enabling a model of stronger generalization.

### 4.4 QUALITATIVE RESULTS AND APPLICATION

Finally, we present qualitative results in Figure 7, demonstrating QueryMeld-Net's strong performance across various segmentation tasks. A notable application of QueryMeldNet is show-cased in image matting, as illustrated in Figure 7(d). Most current image matting methods are class-agnostic Li et al. (2023b; 2021); Liu et al. (2024), which means they do not allow control over which object is segmented. However, with QueryMeldNet, we integrate a refinement module based on AEMatter Liu et al. (2024), enabling controllable image matting. This marriage allows QueryMeldNet to refine instance segmentation to a more precise level, capturing intricate details such as the fur of a horse and the hair of a woman.

Table 3: The comparison to state of the arts on open-set benchmarks. '−' represents no results reported in the original paper. We bold the best entry in each column. For ADE, we report the average number of PQ, mask AP and mIoU for panoptic/instance/semantic segmentation.

| Method | ADE Avg. | PC-59 mIoU | PC-459 mIoU | BDD PQ | SeginW Mask AP |
|---|---|---|---|---|---|
| LSeg+ Ghiasi et al. (2022) | - | 46.5 | 7.8 | - | - |
| SPNet Xian et al. (2019) | - | 24.3 | - | - | - |
| ZS3Net Bucher et al. (2019) | - | 19.4 | - | - | - |
| MaskCLIP Ding et al. (2022) | 14.9 | 45.9 | 10.0 | - | - |
| GroupViT Xu et al. (2022) | - | 25.9 | 4.9 | - | - |
| OpenSeg Ghiasi et al. (2022) | - | 42.1 | 9.0 | - | - |
| ODISE Xu et al. (2023) | 20.7 | 57.3 | 14.5 | - | - |
| X-Decoder Zou et al. (2023a) | 20.5 | 64.0 | 16.1 | 17.8 | 32.3 |
| OpenSeeD Zhang et al. (2023) | 19.0 | - | - | 19.4 | 36.1 |
| DaTaSeg Gu et al. (2023) | - | 51.4 | 11.1 | - | - |
| QueryMeldNet | **20.9** | **65.0** | **18.1** | **29.3** | **43.4** |

### 5 CONCLUSION

In this paper, we have introduced QueryMeldNet, a scalable image segmentation model that can be trained on diverse datasets and tasks at scale. Our experiments have validated the effectiveness of QueryMeldNet in improving segmentation performance as data volume and task diversity increase, particularly in open-set and real-world applications. Moreover, we showed that incorporating synthetic data further boosts the model's generalization capabilities while reducing the reliance on expensive human annotations. QueryMeldNet marks a significant step toward universal segmentation models, opening the door for future research to explore even larger and more complex segmentation tasks.

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

## A    APPENDIX

In this supplement, we show some other additional experimental results and details that are not present in the main paper due to the page limitation.

### A.1    SUPPLEMENTARY EXPERIMENTAL RESULTS

#### A.1.1    FULL RESULTS OF TABLE 3

In Section 4.3 of the main paper, in order to investigate the generalization ability of QueryMeldNet for segmentation, we conduct a zero-shot evaluation of our model on the Segmentation in the Wild (SeginW) benchmark Zou et al. (2023a), which comprises 25 datasets, and report the average mAP of all the datasets. In this supplementary material, we report other additional results including median mAP and individual mAP on each dataset. The results detailed in Table 4 show the superiority of QueryMeldNet over X-Decoder Zou et al. (2023a) and OpenSeeD Zhang et al. (2023) across all datasets. This indicates that the importance of scalability across both datasets and tasks in enhancing the generalization ability of models, a capability unique to QueryMeldNet. In Table 5, we present the complete set of results on the ADE dataset.

#### A.1.2    EXPLICIT RESULTS OF FIGURE 6

In Table 6, we report the numerical results used to generate the five subfigures in Figure 6.

#### A.1.3    ABLATION STUDY

**Enhancement by Synthetic Data** Complementing Section 4.2, here we present more results for demonstrating the significance of synthetic data. 30% images are sampled from Objects365 Shao et al. (2019) training set and synthetic mask is generated for each object with Ke et al. (2023). This subset is denoted as "Objects365-syn-m". We jointly train a model on COCO with instance annotation ("COCO ins") and Objects365-syn-m and compare with the baseline trained on "COCO ins" only. As shown in Table 7, the improvement is clear, suggesting the benefit of using synthetic masks.

Similarly, synthetic object captions are generated for all COCO instances, denoted as "COCO-syn". We trained a model jointly on it with RefCOCOg. The comparison in Table 8 with the baseline shows that the improvement is significant (more than 4 points), indicating the benefits of synthetic captions.

**The Impact of Query Numbers** In this section, we ablate the impact of the number of queries. By default, we use mixture of 100 learnable and 300 conditional queries. This setting is derived from MaskDINO, ADE semantic setting of 100 learnable queries and COCO instance setting of 300 conditional queries. It is also the same as OpenSeeD using 100 learnable queries for stuff classes and 300 conditional queries for thing classes. Based on the Base-scale image and text encoder backbones, given different queries, we scale models with the configuration of two tasks and datasets. The training set is the combination of COCO with instance segmentation and ADE with semantic segmentation. In Table 9, we observe that increasing the query number can improve the performance. However, the memory cost also increases considerably. Because such GPU memory cost is not affordable for our team when scaling up to large-scale backbones, in other experiments across the paper, we keep the "100+300" setting consistently. This also enables a fair comparison to other methods.

#### A.1.4    MODEL SIZE AND SPEED COMPARISON

We evaluate the model size in terms of the numbers of parameters (Params) and conduct a speed comparison by reporting frames-per-second (FPS). The speed tests are performed on A100 NVIDIA GPU with 40GB memory by taking the average computing time with batch size 1 on the entire validation set, using Detectron2 Wu et al. (2019). All models listed in Table 10 are characterized by large-scale backbone models. In general, there is no substantial difference in the forward speed across three models. The increase in parameters for both X-Decoder and our QueryMeldNet over

OneFormer is primarily attributed to the introduction of a language encoder, given that they are open-vocabulary models.

## A.2 ADDITIONAL EXPERIMENTAL DETAILS

**Training settings** For the experiments of Section 4.1, we train our model with a batch size of 32. AdamW is used as the optimizer with a base learning rate of 2e-4 for the segmentation encoder and decoder, and 2e-5 , 10 warmup iterations, and a weight decay of 0.05. We decay the learning rate at 0.9 and 0.95 fractions of the total number of training steps by a factor of 10. We train for a total of 50 epochs. On the experiments of Sections 4.2 and 4.3, we follow the same settings but the batch size is scaled up to 128. Swin-Base and CLIP-Base are used for query comparison in Table 2. Their larger-scale variants are used in other sections. The codes and models will be released upon acceptance.

**Datasets** In order to mitigate the data leakage issue, we implement exclusion in our training data. Specifically, for the COCO 2017 training set, examples belonging to RefCOCO, RefCOCO+, Ref-COCOg validation sets are excluded. Conversely, training examples from RefCOCO, RefCOCO+, RefCOCOg that overlap with COCO 2017 validation set are also excluded. Similar exclusion procedures are applied to LVIS training set, removing examples associated with the RefCOCO, Ref-COCO+, RefCOCOg validation sets. Distinct data augmentation strategies are applied based on the type of training data. For instance, semantic and panoptic data, we follow the augmentation strategy of Mask DINO Li et al. (2023a). For referring segmentation data, the augmentation data is the same as instance segmentation but random clip is excluded. For foreground segmentation training data, we follow the data augmentation of InSPyReNet Kim et al. (2022b). Different upsampling ratios for each dataset are applied during joint training, which are maintained as specified in Table 11. In total, the QueryMeldNet is trained on around 2M distinct images examples and 57M mask annotations. It is noted that the comparison in Table 3 is a system-level comparison. The training data varies across each method. For instance, X-decoder Zou et al. (2023a) additionally incorporates image-text corpora in its training process.

## A.3 ETHICAL CONSIDERATIONS

We discuss the ethical considerations from three aspects: **Environmental Impact:** Training QueryMeldNet requires significant computational resources. The environmental impact of such resource-intensive processes should be taken into account, and efforts should be made to develop more energy-efficient algorithms. **Transparency and Explainability:** Like other deep learning models, QueryMeldNet is also considered "black boxes" because it is challenging to understand how they reach specific decisions. Ensuring transparency and explainability is essential to build trust and accountability, especially in applications with significant consequences. **Bias and Fairness:** Like other machine learning models, image segmentation models can be biased based on the data they are trained on. If the training data is not diverse and representative, the model may perform poorly on certain demographics or groups, perpetuating existing biases. However, this problem can be resolved to a certain extent by QueryMeldNet thanks to its versatility of joint training on multiple diverse datasets and tasks.

## A.4 LIMITATIONS

Recently, a newly emerging reasoning segmentation task has been introduced Lai et al. (2023). The task is designed to output a segmentation mask given a complex and implicit query text. For example, given an image with various fruits, the query is "what is the fruit with the most Vitamin C in this image". This task demands a level of reasoning typically handled by multi-modal Large Language Models. Currently, QueryMeldNet does not explicitly support this task. However, addressing this limitation is part of our agenda for future research.

Table 4: Open-set segmentation comparison on the SeginW benchmark. We bold the best entry in each column.

| Model | Med. | Avg. | Air-Par. | Bottles | Br.Tum. | Chicken | Cows | Ele-Sha. | Eleph. | Fruits | Gar. | Gin-Gar. | Hand | Hand-Metal | House-Parts | HH-Items | Nut-Squi. | Phones | Poles | Puppies | Rail | Sal-Fil. | Stra. | Tablets | Toolkits | Trash | W.M |
|---|---|---|---|---|---|---|---|---|---|---|---|---|---|---|---|---|---|---|---|---|---|---|---|---|---|---|---|
| X-Decoder Zou et al. (2023a) | 22.3 | 32.3 | 13.1 | 42.1 | 2.2 | 8.6 | 44.9 | 7.5 | 66.0 | 79.2 | 33.0 | 11.6 | 75.9 | 42.1 | 7.0 | 53.0 | 68.4 | 15.6 | 20.1 | 59.0 | 2.3 | 19.0 | 67.1 | 22.5 | 9.9 | 22.3 | 13.8 |
| OpenSeeD Zhang et al. (2023) | 38.7 | 36.1 | 13.1 | 39.7 | 2.1 | 82.9 | 40.9 | 4.7 | 72.9 | 76.4 | 16.9 | 13.6 | 92.7 | 38.7 | 1.8 | 50.0 | 40.0 | 7.6 | 4.6 | 74.6 | 1.8 | 15.0 | 82.8 | 47.4 | 15.4 | 15.3 | 52.3 |
| QueryMeldNet | **43.0** | **43.4** | **14.4** | **44.4** | **3.3** | **85.2** | **45.0** | **15.0** | **75.2** | **80.4** | **33.1** | **20.9** | **94.4** | **44.6** | **7.8** | **54.2** | **69.5** | **16.0** | **24.2** | **78.0** | **4.4** | **27.8** | **84.5** | **49.3** | **23.2** | **35.5** | **59.4** |

Table 5: The comparison to state of the arts on open-set benchmarks. '−' represents no results reported in the original paper. We bold the best entry in each column.

| Method | ADE | | | | PC-59 mIoU | PC-459 mIoU | BDD PQ | SeginW Mask AP |
| | PQ | Mask AP | Box AP | mIoU | | | | |
|---|---|---|---|---|---|---|---|---|
| LSeg+ Ghiasi et al. (2022) | - | - | - | 18.0 | 46.5 | 7.8 | - | - |
| MSeg Lambert et al. (2020) | - | - | - | 19.1 | - | - | - | - |
| SPNet Xian et al. (2019) | - | - | - | - | 24.3 | - | - | - |
| ZS3Net Bucher et al. (2019) | - | - | - | - | 19.4 | - | - | - |
| MaskCLIP Ding et al. (2022) | 15.1 | 6.0 | 14.9 | 23.7 | 45.9 | 10.0 | - | - |
| GroupViT Xu et al. (2022) | - | - | - | 10.6 | 25.9 | 4.9 | - | - |
| OpenSeg Ghiasi et al. (2022) | - | - | - | 21.1 | 42.1 | 9.0 | - | - |
| ODISE Xu et al. (2023) | **22.6** | 14.4 | 15.8 | **29.9** | 57.3 | 14.5 | - | - |
| X-Decoder Zou et al. (2023a) | 21.8 | 13.1 | 17.5 | 29.6 | 64.0 | 16.1 | 17.8 | 32.3 |
| OpenSeeD Zhang et al. (2023) | 19.7 | 15.0 | 17.7 | 23.4 | - | - | 19.4 | 36.1 |
| DaTaSeg Gu et al. (2023) | - | - | - | - | 51.4 | 11.1 | - | - |
| QueryMeldNet | 22.1 | **17.3** | **19.2** | 25.0 | **65.0** | **18.1** | **29.3** | **43.4** |

Table 6: The performance improvement with data and task scaling up.

**Subfigure 1**

| Data | | Task | | Referring segmentation |
| Dataset | Size (M) | Type | Number | RefCOCOg (mIoU) |
| --- | --- | --- | --- | --- |
| RefCOCO,RefCOCO+,RefCOCOg | 0.06 | Referring segmentation | 1 | 57.8 |
| RefCOCO,RefCOCO+,RefCOCOg, COCO-syn | 0.16 | Referring segmentation | 1 | 60.8 |
| RefCOCO,RefCOCO+,RefCOCOg, COCO-syn, 30% Objects365-syn | 0.67 | Referring segmentation | 1 | 62.6 |

**Subfigure 2**

| Data | | Task | | Open-vocabulary segmentation |
| Dataset | Size (M) | Type | Number | SeginW (Mask AP) |
| --- | --- | --- | --- | --- |
| COCO | 0.1 | Instance segmentation | 1 | 29.4 |
| COCO, ADE20K | 0.12 | Instance segmentation | 1 | 30.0 |
| COCO, ADE20K, 30% Objects365-syn-m | 0.63 | Instance segmentation | 1 | 35.5 |

**Subfigure 3**

| Data | | Task | | Open-vocabulary segmentation |
| Dataset | Size (M) | Type | Number | SeginW (Mask AP) |
| --- | --- | --- | --- | --- |
| COCO | 0.1 | Panoptic segmentation | 1 | 29.6 |
| COCO | 0.1 | Panoptic, instance segmentation | 2 | 32.7 |
| COCO | 0.1 | Panoptic, instance, referring segmentation | 3 | 35.2 |

**Subfigure 4**

| Data | | Task | | Referring segmentation |
| Dataset | Size (M) | Type | Number | RefCOCOg (mIoU) |
| --- | --- | --- | --- | --- |
| COCO | 0.1 | Panoptic, referring segmentation | 2 | 63.4 |
| COCO, ADE20K, RefCOCO,RefCOCO+,RefCOCOg, LVIS, VG, COCO-syn | 0.3 | Panoptic, instance, referring segmentation | 3 | 64.3 |
| COCO, ADE20K, RefCOCO,RefCOCO+,RefCOCOg, VG, fore, LVIS, Objects365-syn | 2.2 | Panoptic, instance, semantic, referring, foreground segmentation, object detection | 6 | 67.2 |

**Subfigure 5**

| Data | | Task | | Open-vocabulary segmentation |
| Dataset | Size (M) | Type | Number | SeginW (Mask AP) |
| --- | --- | --- | --- | --- |
| COCO | 0.1 | Panoptic, referring segmentation | 2 | 33.2 |
| COCO, ADE20K, RefCOCO,RefCOCO+,RefCOCOg, VG, fore | 0.4 | Panoptic, instance, referring, foreground segmentation | 4 | 37.2 |
| COCO, ADE20K, RefCOCO,RefCOCO+,RefCOCOg, VG, fore, 30% Objects365-syn | 1.0 | Panoptic, instance, referring, foreground segmentation | 4 | 39.6 |
| COCO, ADE20K, RefCOCO,RefCOCO+,RefCOCOg, VG, fore, LVIS, Objects365-syn | 2.2 | Panoptic, instance, semantic, referring, foreground segmentation, object detection | 6 | 43.4 |

Table 7: The impact of synthetic masks

| Training data | Mask AP | Box AP |
|---|---|---|
| COCO ins | 49.7 | 55.3 |
| COCO ins + Objects365-syn-m | 50.5 | 56.8 |

Table 8: The impact of synthetic captions

| Training data | mIoU |
|---|---|
| RefCOCOg | 57.8 |
| syn-COCO | 58.8 |
| RefCOCOg + COCO-syn | 62.6 |

Table 9: The impact of query numbers.

| #learnable+#conditional | ADE | COCO | |
|---|---|---|---|
| | mIoU | Mask AP | Box AP |
| 100+300 | 51.7 | 49.6 | 54.9 |
| 300+900 | 52.0 | 50.7 | 57.4 |

Table 10: The model size and speed comparison.

| Method | Params | FPS |
|---|---|---|
| OneFormer Jain et al. (2023) | 219M | 5.6 |
| X-Decoder Zou et al. (2023a) | 280M | 6.1 |
| QueryMeldNet | 286M | 5.1 |

Table 11: Upsampling ratio of joint training. "referring" refers to the combination of RefCOCO, RefCOCO+, RefCOCOg Kazemzadeh et al. (2014); Yu et al. (2016). "foreground" refers to the combination of seven foreground datasets, HRSOD Zeng et al. (2019), DIS Qin et al. (2022), THUS Cheng et al. (2015), COIFT Mansilla & Miranda (2016), ThinObjects5K Liew et al. (2021), UHRSD Xie et al. (2022), DUTS Wang et al. (2017).

| Dataset | Ratio | #Images | #Annotations |
|---|---|---|---|
| COCO | 3 | 100K | 1.3M |
| ADE20K | 30 | 20K | 271K |
| LVIS | 3 | 100K | 1.3M |
| Visual Genome | 9 | 100K | 2.3M |
| Objects365 | 1 | 1.7M | 25M |
| referring | 6 | 54K | 124K |
| syn-COCO | 3 | 100K | 1.3M |
| syn-Objects365 | 1 | 1.7M | 25M |
| foreground | 9 | 100K | 100K |

