# OpenReview forum: "Scaling up Image Segmentation across Data and Tasks"
_ICLR.cc/2025/Conference — ICLR 2025 Conference Withdrawn Submission_

### Official Review · Reviewer_SUct · 2024-10-30

**Soundness:** 3
**Presentation:** 3
**Contribution:** 3
**Rating:** 5
**Confidence:** 4

**Summary:**

The paper introduces QueryMeldNet, a scalable image segmentation framework designed to improve generalization across diverse datasets and tasks. It proposes a novel object query mechanism called "query meld," which combines learnable and conditional queries using cross-attention. This approach allows the model to balance instance- and stuff-level segmentation, enhancing its scalability. By training on multiple datasets and tasks, QueryMeldNet continuously improves performance, particularly in open-set segmentation tasks. The inclusion of synthetic data further boosts the model's generalization capabilities, reducing the need for costly human annotations. The paper demonstrates that scaling up both the volume and diversity of data and tasks consistently enhances the model's segmentation capabilities, marking a significant step toward universal, scalable segmentation models.

**Strengths:**

The paper introduces QueryMeldNet, a novel segmentation framework that addresses the limitations of existing task-specific models by proposing a scalable solution that integrates knowledge across diverse datasets and tasks. The results of this paper show consistent improvements in segmentation performance as the volume and diversity of data and tasks increase, particularly in open-set segmentation tasks.

The paper is well-written and clear in its presentation. The introduction provides a comprehensive overview of the problem and the motivation for the proposed solution. The methodology section is detailed and easy to follow, with clear explanations of the query meld mechanism and how it integrates with the overall architecture.

**Weaknesses:**

* The expression of Fig.2 is unclear, and I cannot intuitively understand the roles of text prompts and encoders from Fig.2.
* Fig.6 is too small. I can hardly see the text above.
* The performance improvement drive from your proposed query meld mechanism or multiple data co-training is not very clearly compared.  Is the performance improvement due to your use of more synthetic data
* The paper should validate the model's performance on more diverse datasets that are not part of the training set. This would demonstrate the true generalization capability of QueryMeldNet in real-world scenarios.
* The paper should provide more details on the synthetic data generation process, including the models used (e.g., SAM, OFA) and the quality of the generated data.

**Questions:**

How do you ensure that the synthetic data complements the real data effectively? Are there any strategies to balance the use of synthetic and real data during training?

---

### Official Review · Reviewer_biAR · 2024-11-02

**Soundness:** 2
**Presentation:** 3
**Contribution:** 2
**Rating:** 3
**Confidence:** 4

**Summary:**

In this paper, the authors propose a method to train a model that can scale up image segmentation across different tasks and data. To achieve this goal, the authors propose a dynamic object query scheme and a synthetic data generation method.

**Strengths:**

1. The goal of this paper is reasonable. It would help downstream tasks if we could have a strong generalized segmentation model.
2. The authors conduct ablation studies to show the effectiveness of the proposed method.
3. The authors show some visualization results to demonstrate the effectiveness.

**Weaknesses:**

1. The paper lacks system-level performance comparison with previous important baselines. In Table 2, the authors do the ablation experiments about the query strategies. However, the authors do not provide a detailed comparison with previous baselines (as shown in Table 2). Even the comparisons of COCO instance and panoptic segmentation, two of the most important benchmarks in image segmentation, are not given. Also, some recent works [R1, R2] were not compared in the table. In fact, the compared works were all online before mid-2023. Nevertheless, the proposed method does not have a notable better performance on main tasks. This will drastically trivialize the “scaling up” since we hope to see notable results after “scaling up”


2. In Table 1, the authors compare the supported tasks with previous works. Based on the comparison, the proposed method supports extra foreground segmentation.  However, the experiments do not show the effectiveness of “foreground segmentation.” In addition, I do not think supporting “foreground segmentation” is a significant contribution, although the authors highlight that.

3. The technical contribution of the module design is not well studied. The authors want to claim that the different types of queries will have different specialties so that a combination of them will have a better result. However, the ratio of these two types of queries has not been studied in detail. Also, given that in Mask2former (a typical learnable query method), if the query number is too much, the performance may drop. Setting the number of queries to 400 may not be a good choice for fair comparison. At the same time, since the performance gain is not very obvious, it is questionable whether similar conclusions can be drawn with large amounts of data. Also, there are some other papers, such as Oneformer, that tackle semantic/instance joint training issue. Comparing these papers under the same setting can also help better understand the effectiveness of the method. In conclusion, I think these two types of queries are not new methods, and their combination has not been fully studied in this paper.

4. The method and contribution of data scaling up seem not significant. Mask labeling object365 is not new, and [R3] provides a much larger object-level captioning dataset. Therefore, both the “scaling data” and the effect of that are not notable.



[R1] Hierarchical Open-vocabulary Universal Image Segmentation.

[R2] OMG-Seg: Is One Model Good Enough for All Segmentation?

[R3] Tokenize anything via prompting.

**Questions:**

5. (question) L302.  Based on my understanding, it is hard to consider the window in the stuff class and the window in the thing class as the same category because the former tends to segment all windows while the latter tends to segment one window. Considering this, is the implementation of OpenSeeD more reasonable?

---

### Official Review · Reviewer_NUHH · 2024-11-03

**Soundness:** 2
**Presentation:** 1
**Contribution:** 3
**Rating:** 5
**Confidence:** 4

**Summary:**

This paper presents QueryMeldNet, an image segmentation model that is designed to be trained *at scale*, i.e., on many different datasets and segmentation tasks. QueryMeldNet is a query-based segmentation model, which makes segmentation predictions with a Transformer-based segmentation decoder that takes image features and a set of queries as inputs. In QueryMeldNet, this set of queries consists of both learnable queries and conditional queries (generated by an encoder). This is empirically shown to lead to a better segmentation performance than using just one of these query types, as done by existing models. With experiments on many datasets and segmentation tasks, QueryMeldNet is shown to scale effectively with the dataset volume and number of training tasks, i.e., the segmentation performance increases when the model is trained on more data or more segmentation tasks. Additionally, the paper shows that QueryMeldNet can also effectively be trained on images with synthetically generated labels, further improving the performance.

**Strengths:**

1.	The paper tackles an interesting and relevant problem, i.e., training a segmentation model on multiple datasets with annotations for different segmentation tasks, to obtain a segmentation model that generalizes well. Most existing segmentation models cannot be trained on multiple datasets with different label definitions. As a result, they cannot leverage the complementary information that is contained within different datasets, potentially limiting their accuracy and generalizability. By training on multiple datasets, this potential could be unlocked.
2.	The method proposed to solve the aforementioned problem, called QueryMeldNet, is experimentally shown to be effective. As it is trained on more data and annotations for more segmentation tasks, QueryMeldNet becomes more accurate and achieves better generalization. Moreover, this behavior persists when it is trained on dataset with synthetic labels. .
3.	The main technical contribution, i.e., the use of both learnable and conditional queries, called ‘query meld’, is innovative and experimentally shown to be effective. In Tab. 2, it is shown that using ‘query meld’ allows the model to consistently obtain the best results across different evaluation tasks, whereas using only the learnable or conditional queries always leads to a suboptimal performance on at least one task.
4.	In Sec. 4.2, the paper conducts insightful experiments by separately evaluating the impact of (a) scaling up the data volume and (b) scaling up the number of tasks for which there are annotations. The results show that it isn’t only beneficial to train on more data, but that the model also generalizes better when it is trained on annotations for multiple tasks, i.e., panoptic segmentation, instance segmentation, and referring segmentation in this case. These results show that it is valuable to scale up a model not only by training it on more data, but also on more diverse annotations.

**Weaknesses:**

1.	While the results show that QueryMeldNet is effective, large parts of the operation of the method aren’t explained properly. As a result, it is not fully clear how the method works, and why it behaves the way it does. The paper would be significantly better if the method were explained more carefully and in more detail. Concretely:

  1.	In L243, the paper mentions that the learnable and conditional queries interact with each other via “dual-query cross-attention”. As such, it is a major component of the newly proposed ‘query meld’ strategy. However, the paper does not provide any details about this cross-attention, making it unclear how it works, and how these queries interact. Do both types of queries cross-attend to each other? If so, in what order? What exact operations are used? These details are necessary to be able to understand and reproduce the model.

  2.	Similarly, the paper does not explain how the conditional queries are generated. L222-L225 describes that these queries are derived from the transformer encoder, and that this encoder is trained to predict region proposals, but no further details are provided. How does the encoder generate these queries exactly? Does QueryMeldNet use an existing method to generate these, or a new design? Is this ‘transformer encoder’ the same encoder as the ‘segmentation encoder’ shown in Fig. 2, or is it a separate component? The answers to these questions should be available in the manuscript.

  3.	In general, it is not clear what the architectural design is of the individual components of QueryMeldNet as shown in Fig. 2. The Appendix mentions that the model uses a Swin image encoder and CLIP text encoder, but the architecture of the segmentation encoder and segmentation decoder are not described. To understand the model and for reproducibility, this information should be available.

  4.	L299 mentions that, unlike OpenSeeD, the proposed QueryMeldNet method does not require that it is defined if a certain class is a ‘thing’ or ‘stuff’ class, i.e., if it requires a mask per individual object or a mask for all objects of a particular class together, respectively. However, the paper does not describe how QueryMeldNet handles situations where the ‘table’ is a thing class in one training dataset but a stuff class in another (as mentioned in L301, this is happens for ADE20K and COCO). How does the model handle these conflicting supervision signals? This is currently not clear, but it should be explained in the paper.

  5.	Similarly, it is not clear how the model knows what task it should conduct. For instance, for the results in Tab. 2, how does the model know that it should conduct semantic segmentation for ADE20K (and predict a mask for all cars together), but panoptic segmentation for COCO (and predict a mask for each individual car)? Is there an additional input to the model that specifies which task it should do? The paper should clearly describe how the model handles this.

2.	In Tab. 3, the proposed QueryMeldNet paper achieves the best reported results on all benchmarks, including the SeginW benchmark for ‘in-the-wild’ segmentation. This makes it seem like QueryMeldNet achieves state-of-the-art results on this dataset. However, Grounded-HQ-SAM [a] achieves a mean AP of 49.6 for this dataset, compared to 43.4 by QueryMeldNet, showing that QueryMeldNet is far from the state of the art. To fairly compare QueryMeldNet to prior works, the performance of Grounded-HQ-SAM should be reported in Tab. 3, or the paper should clearly explain why such a comparison is not necessary.
3.	The presentation of some figures should be improved.

  1.	In Fig. 1 (right), the linear line that interpolates between the different scores of QueryMeldNet is not meaningful. The x-axis is not linear and each point on the x-axis changes more than one variable (the data volume and number of tasks), so the linear interpolation between these results is not appropriate. The plot would improve if this line were removed.

  2. Fig. 1 (left) is not clear. What does the dotted rectangle represent? And what do the arrows from the datasets outside of this dotted rectangle mean? What is the difference between the datasets inside and outside the rectangle? This is not clear. The figure should be clarified, or there should be additional explanations in the caption.
  3.	In general, the plots in Fig. 1 (right) and  Fig. 6 are so small that they are not properly readable when the paper is printed. Moreover, the text in the plots (‘QueryMeldNet’ and ‘X-Decoder’) overlaps with the y-axis ticks and labels. The readability of the figures would improve if the plots (or at least the font size) were bigger, and if the text didn’t overlap.
4.	In Fig. 6 (right), QueryMeldNet is compared to OpenSeeD, and QueryMeldNet is shown to achieve a better performance. However, in this comparison, OpenSeeD and QueryMeldNet are trained on different datasets. Therefore, from this experiment, it is not clear what portion of the performance improvement is due to the quality of the datasets used for training, and what portion is due to the difference in model architecture or training strategy. To get insights in this manner, and improve the value of this comparison, an experiment could be conducted where QueryMeldNet is trained on the same 1.8M data samples as OpenSeeD, across the same 2 tasks.
5.	In L133, the paper states that QueryMeldNet can be trained and evaluated on **any** segmentation task. However, there are several segmentation tasks for which it is not clear how QueryMeldNet would solve them, such as part-aware panoptic segmentation [b] which requires objects and their parts to be segmented jointly, or promptable segmentation like SAM [c] does. Could QueryMeldNet solve these tasks? If so, how? If not, then the proposed method would still be valuable, of course, but then the statement in L133 should be changed.
6.	Tab. 1 shows that X-Decoder [d] cannot conduct foreground segmentation and object detection, while the newly proposed QueryMeldNet can. However, the paper does not explain what changes it makes (compared to X-Decoder) to enable foreground segmentation and object detection. Besides the additional use of conditional queries, are there any fundamental differences between X-Decoder and QueryMeldNet? Or is X-Decoder just not trained for foreground segmentation and object detection, while it could have been trained for that purpose without significant changes? If the paper explained this, the value of QueryMeldNet over X-Decoder would be much clearer.
7.	In L252-L258, the paper describes that a benefit of the ‘query meld’ strategy is that the two types of queries can lead to a “hierarchical and interactive feature representation,” where learnable queries capture the overall structure of the scene and conditional queries refine these global queries by attending to specific parts of the image. However, the paper does not provide any evidence that this happens. Could the authors provide evidence to support this claim? If not, then it should be clear that this is something that the authors expect might happen in the model, and not a certainty.
8.	There are several text-related mistakes, limiting the readability. Some examples:

  1.	The citations are not in the proper format. For instance, L221: “two-stage object detection framework Ren et al. (2015)” should be “two-stage object detection framework (Ren et al., 2015). To fix this, the authors should use the `\citep{ref}` command, instead of `\citet{ref}`. Sec. 4.1 of the ICLR 2025 template provides more detailed instructions.

  2.	L022-L023: the hyphens in “data-generating” and “tasks-drastically” should be ‘en dashes’ or ‘em dashes’.

  3.	L090: “and hierarchical and interactive feature representation” => “and a hierarchical and interactive feature representation” or “and hierarchical and interactive feature representations”

  4.	L267: “This thanks to” => “This is thanks to”

  5.	L314-L315: “Since all … Figure 2.” is not a proper sentence. It could be improved in various ways, e.g., by removing the word “Since”.

  6.	L269: “is a triplet depicts” => “is a triplet that depicts”

  7.	L316: “dealing open-vocabulary” => “dealing with open-vocabulary”

  8.	L318: “lead QueryMeldNet is scalable” could be improved by something like “make QueryMeldNet scalable”

  9.	L456: “each tasks” => “each task”

   10.	The notation in Eq. 1 is not clear. Are $\textbf{P}^c$, $\textbf{P}^b$ and $\textbf{P}^m$ tensors, as suggested by the fact that the text (L310) calls them embeddings? Or are they functions, as suggested by the fact that they seem to take $\textbf{x}_i$ as an input in Eq. 1? This is confusing, and the notation should be improved.

[a] Ke et al., "Segment Anything in High Quality," NeurIPS 2023.

[b] De Geus et al., “Part-aware Panoptic Segmentation,” CVPR 2021.

[c] Kirillov et al., “Segment Anything,” ICCV 2023.

[d] Zou et al., “Generalized Decoding for Pixel, Image, and Language,” CVPR 2023.

**Questions:**

The main reasons for the low rating are the inadequate explanations of QueryMeldNet's operation, the missing comparison on the SGinW benchmark, the unjustified claims, and the overall presentation considering the figures and the text (see the ‘weaknesses’ section). I would like to ask the authors to carefully address my concerns, answer the questions posed in the ‘weaknesses’ section, and revise the manuscript accordingly.

Additionally, I have two more questions:

1.	In Tab. 2, the ‘query meld’ strategy is shown to be significantly better than the ‘learnable query’ or ‘conditional query’ strategy. However, after reading the paper, it is not clear to me why this problem is specific to training a segmentation model on multiple datasets and tasks. Does the problem with learnable queries also occur for ‘regular’ panoptic segmentation models (e.g., Mask2Former [e] on COCO), as they also deal with both stuff and thing classes? If so, the paper could be made even stronger by showing that ‘query meld’ is also an effective strategy for these existing models.
2.	In Fig. 6 (middle) and Tab. 6, the SeginW performance is shown to improve significantly when training not only on panoptic segmentation labels, but also on instance segmentation labels. How is this possible? The instance segmentation annotations are already contained in the panoptic segmentation annotations (they are simply the ‘things’ annotations), so why does the performance increase when additionally training on these labels?

[e] Cheng et al., “Masked-attention Mask Transformer for Universal Image Segmentation,” CVPR 2022.

---

### Official Review · Reviewer_3jmd · 2024-11-03

**Soundness:** 3
**Presentation:** 3
**Contribution:** 2
**Rating:** 6
**Confidence:** 4

**Summary:**

The paper proposes to combine conditional query tokens and learnable query tokens for better open-world image segmentation. It scales up the existing datasets and tasks utilized to train the segmentation model. Beyond this, it proposes a data engine to craft segmentation masks from existing large-scale object detection tasks, such as Objects365. With the significantly scaled-up data, the model performs evidently better than previous methods.

**Strengths:**

1. The paper is well-written and very easy to follow. The main figure clearly conveys the core structural contribution of this work. Moreover, the analysis before presenting the concrete technical modification is insightful.

2. Although the model modification is very simple (combining two forms of query tokens), it is well-motivated and highly effective. I appreciate such a simple yet effective design very much.

3. The data engine part is also well-motivated. Existing segmentation datasets are indeed relatively small. It is insightful to seek additional detection datasets to construct high-quality segmentation data.

**Weaknesses:**

I think if the position of this paper is to push the boundaries of all existing open-world segmentor, it would be better to provide different scales of pre-trained models for the community to use, *e.g.*, based on ViT-S, ViT-B, and ViT-L. Moreover, it will be very good to see better results under more advanced vision encoders than Swin Transformers.

**Questions:**

How about the results under more advanced vision encoders like DINOv2? Could the authors provide such additional results?

---

### Official Review · Reviewer_d3TA · 2024-11-04

**Soundness:** 3
**Presentation:** 3
**Contribution:** 2
**Rating:** 5
**Confidence:** 3

**Summary:**

This paper presents an innovative image segmentation technology. First, this paper points out the key limitation of the object query design in the Transformer-based segmentation model. It proposes a new object query mechanism called “query meld”. This mechanism combines learnable queries and conditional queries to achieve dynamic query selection and hierarchical and interactive feature representation, thereby enhancing the model's ability to handle different object types. Based on this, a scalable segmentation architecture named QueryMeldNet is constructed. It can be trained on multiple segmentation tasks and datasets. With the increase in the amount of training data and the diversity of tasks, the segmentation performance is continuously improved, especially in real-world, free-form open-set segmentation tasks. In addition, synthetic data is used to alleviate the problem of data scarcity, enhance the model's robustness and semantic understanding, and further improve the performance of QueryMeldNet.

**Strengths:**

1.	The authors propose a scalable segmentation architecture. By introducing a new object query mechanism "query meld" that combines learnable queries and conditional queries, the model is enabled to handle various types of objects, and its adaptability across different tasks and datasets is enhanced.
2.	In response to the high cost of annotating segmentation data, authors introduce synthetic data to scale up training. This approach alleviates the problem of data scarcity and enhances the model's robustness and generalization ability.
3.	The authors prove that expanding the training of the model across diverse tasks and datasets can continuously improve its generalization ability, especially performing outstandingly in real-world, free-form open-set segmentation tasks, breaking the constraints of task-specific or dataset-specific models.
4.	Through the above innovations, QueryMeldNet achieves state-of-the-art performance in multiple open-set segmentation benchmarks.

**Weaknesses:**

1.	What is the specific structure of the “dual-query cross-attention”? Is it simply directly concatenating the condition queries and learnable queries together and then using them as the query of vanilla cross-attention? This part requires an explanation.
2.	The author needs to explain the difference between the proposed query design and the query of OpenSeeD[1], considering both selecting queries from feature + learnable query.
3.	Using synthetic data to enhance training is one of the main contributions. Will the code and synthetic dataset be open-sourced?
4.	Minor writing error: In Figure 3(c), should the output of the decoder be circled with slashes instead of squares with slashes?

Some related works are missing:

[a] COCONut: Modernizing COCO Segmentation https://arxiv.org/pdf/2404.08639

[b] Convolutions Die Hard: Open-Vocabulary Segmentation with Single Frozen Convolutional CLIP https://arxiv.org/pdf/2308.02487

[c] OMG-Seg: Is One Model Good Enough For All Segmentation? https://arxiv.org/abs/2401.10229

[d] Open-Vocabulary SAM: Segment and Recognize Twenty-thousand Classes Interactively https://arxiv.org/pdf/2401.02955

**Questions:**

See above weaknesses.

---

### Author Response · Authors · 2024-11-15

We sincerely thank the ACs, reviewers, and PCs for their time and thoughtful feedback, which greatly enhanced our work. After careful consideration, we have decided to withdraw our paper, but we deeply appreciate the support and constructive input.

---

### Note · Authors · 2024-11-15

I have read and agree with the venue's withdrawal policy on behalf of myself and my co-authors.